# Clinicopathological Impact of the Spread through Air Space in Non-Small Cell Lung Cancer: A Meta-Analysis

**DOI:** 10.3390/diagnostics12051112

**Published:** 2022-04-28

**Authors:** Jung-Soo Pyo, Nae Yu Kim

**Affiliations:** 1Department of Pathology, Uijeongbu Eulji Medical Center, Eulji University School of Medicine, Uijeongbu-si 11759, Gyeonggi-do, Korea; jspyo@eulji.ac.kr; 2Department of Internal Medicine, Uijeongbu Eulji Medical Center, Eulji University School of Medicine, Uijeongbu-si 11759, Gyeonggi-do, Korea

**Keywords:** lung adenocarcinoma, lung squamous cell carcinoma, spread through air space, growth pattern, sublobar resection, prognosis, meta-analysis

## Abstract

This study aimed to elucidate the clinicopathological significance of spread through air space (STAS) in non-small cell lung cancer (NSCLC) through a meta-analysis. Using 47 eligible studies, we obtained the estimated rates of STAS in various histological subtypes of NSCLC and compared the clinicopathological characteristics and prognosis between NSCLC with and without STAS. The estimated STAS rate was 0.368 (95% confidence interval [CI], 0.336–0.0.401) in patients with NSCLC. Furthermore, the STAS rates for squamous cell carcinoma and adenocarcinoma were 0.338 (95% CI, 0.273–0.411) and 0.374 (95% CI, 0.340–0.409), respectively. Among the histological subtypes of adenocarcinoma, micropapillary-predominant tumors had the highest rate of STAS (0.719; 95% CI, 0.652–0.778). The STAS rates of solid- and papillary-predominant adenocarcinoma were 0.567 (95% CI, 0.478–0.652) and 0.446 (95% CI, 0.392–0.501), respectively. NSCLCs with STAS showed a higher visceral pleural, venous, and lymphatic invasion than those without STAS. In addition, anaplastic lymphoma kinase mutations and ROS1 rearrangements were significantly more frequent in NSCLCs with STAS than in those without STAS. The presence of STAS was significantly correlated with worse overall and recurrence-free survival (hazard ratio, 2.119; 95% CI, 1.811–2.480 and 2.372; 95% CI, 2.018–2.788, respectively). Taken together, the presence of STAS is useful in predicting the clinicopathological significance and prognosis of patients with NSCLC.

## 1. Introduction

Lung cancer is one of the most common causes of cancer-related deaths worldwide [1]. In the recent treatment of lung cancer, histological classification, including molecular and biomarker profiles, is important due to the need to decide on systemic therapies [1]. Kadota et al. introduced “spread through airspace (STAS)” in lung tumors [2]. STAS is defined as the spread of lung cancer cells into the air spaces adjacent to the main tumor [2]. STAS should be distinguished from the artificial spreading features. For example, the discontinuity of spread in airspace from the tumor edge is ruled out as an artifact [2]. In addition, in daily practice, the differentiation of spreading tumor cells from normal, benign pneumocytes or bronchial cells can be difficult [2]. They described three morphological patterns (micropapillary structures, solid nests of tumor cells, and discohesive single cells) which are frequently identified in STAS of adenocarcinoma (ADC) [2]. STAS rates according to the histological subtypes of ADC can be different. Non-small cell lung cancer (NSCLC) includes ADC, squamous cell carcinoma (SCC), and large cell carcinoma. Adenocarcinomas contain various histological subtypes, such as lepidic, acinar, papillary, micropapillary, and solid [1]. STAS was significantly correlated with lymphatic and vascular invasions [2]. In addition, STAS was frequently found in lung cancer with papillary, micropapillary, and solid patterns [2]. However, STAS was less frequent in lung cancer with a lepidic pattern than in those without a lepidic pattern [2]. However, although previous studies have reported the prognostic roles of STAS, detailed information on STAS rates according to histological subtypes is unclear [2,3,4,5,6]. Surgical specimens from limited resection can be limited in the evaluation of STAS due to the limitation of the adjacent parenchyma [2,4]. In addition, because STAS does not include the measurement of tumor size, there is no impact of STAS on tumor staging. Therefore, due to the possibility of understaging, further evaluation of the impact of STAS on tumor stage is needed. The clinicopathological implications of the presence of STAS can differ between patients with the same histological subtypes and tumor staging. The correlation between histological subtypes and STAS may be more important. However, detailed information based on histological subtypes is unclear. This study aimed to elucidate the clinicopathological significance of STAS in NSCLC through a meta-analysis. First, the estimated rates of STAS were investigated and evaluated in various histological subtypes and clinicopathological subgroups. In addition, the prognostic implications of STAS were investigated, and a subgroup analysis was performed.

## 2. Materials and Methods

### 2.1. Published Studies Search and Selection Criteria

Searching was performed using the PubMed and MEDLINE databases on 30 June 2021. These databases were searched using the following keywords: “lung” and “STAS or spread through air spaces.” The titles and abstracts of all searched articles from databases were screened for exclusion. Review articles were also screened to find additional eligible studies. Articles were included if the study was performed in human NSCLC and if there was information about the clinicopathological characteristics and prognosis of NSCLC with and without STAS. Articles were excluded if they were case reports or non-original articles or if the article was not written in English.

### 2.2. Data Extraction

The data was extracted from each of the eligible studies by two researchers [2,4,7,8,9,10,11,12,13,14,15,16,17,18,19,20,21,22,23,24,25,26,27,28,29,30,31,32,33,34,35,36,37,38,39,40,41,42,43,44,45,46,47,48,49,50,51]. Extracted data included: the first author’s name, year of publication, study location, number of patients analyzed, and clinicopathological information for patients with and without STAS. Investigated clinicopathological information included histologic subtypes, patients’ age and sex, smoking history, tumor size, tumor location, visceral pleural, venous, and lymphatic invasion, genetic mutations of *anaplastic lymphoma kinase (ALK), epithelial growth factor receptor (EGFR), ROS1,* and *KRAS*, and survival rate.

### 2.3. Statistical Analyses

To perform the meta-analysis, all data were analyzed using the Comprehensive Meta-Analysis software package (Biostat, Englewood, NJ, USA). The incidence rates of STAS were investigated in NSCLCs. In addition, the presence of STAS and various clinicopathological characteristics, including genetic mutations, were investigated and performed in the meta-analysis. The correlations between the presence of STAS and overall and recurrence-free survivals were evaluated. For a quantitative aggregation of survival results, we obtained the hazard ratio (HR) using one of the following methods. In studies not quoting the HR or its confidence interval (CI), these variables were calculated from the presented data using the HR point estimate, log-rank statistic or its *p*-value, and the O-E statistic (difference between the number of observed and expected events) or its variance. If those data were unavailable, HR was estimated using the total number of events, number of patients at risk in each group, and the log-rank statistic or its *p*-value. Finally, if the only useful data were in the form of graphical representations of survival distributions, survival rates were extracted at specified times to reconstruct the HR estimate and its variance under the assumption that patients were censored at a constant rate during the time intervals [52]. The published survival curves were read independently by two researchers in order to reduce variability. The HRs were then combined into an overall HR using Peto’s method [53]. Heterogeneity between the studies was checked by the Q and I^2^ statistics and expressed as *p*-values. Additionally, sensitivity analysis was conducted to assess the heterogeneity of eligible studies and the impact of each study on the combined effect. In addition, to compare between subgroups with and without STAS, the meta-regression test was performed. Because eligible studies used various populations, the application of the random-effect model rather than the fixed-effect model was more suitable. For the assessment of publication bias, Begg’s funnel plot and Egger’s test were used. If significant publication bias was found, the fail-safe N and trim-fill tests were additionally conducted to confirm the degree of publication bias. The results were considered statistically significant at *p* < 0.05.

## 3. Results

### 3.1. Selection and Characteristics of the Studies

In this study, 47 studies were included among the 201 searched studies. In total, 51 studies were excluded because they were non-original articles. Moreover, 46 articles had insufficient or no information. Overall, 44 articles were studied for other diseases. Two reports were excluded due to duplication of patients. In addition, 11 reports were excluded due to non-English (*n* = 8) and non-human samples (*n* = 3). Detailed information for the included and excluded studies is presented in Figure 1 and Table 1.

### 3.2. Estimated Rates of STAS in NSCLC

The estimated rate of STAS was 0.368 (95% CI, 0.336–0.401) in patients with NSCLC (Table 2). STAS was found in 33.8% and 37.4% of the cases of SCC and ADC, respectively. In the subgroup analysis based on histological subtypes of ADC, the STAS rate was the highest in micropapillary-predominant ADC (0.719; 95% CI, 0.652–0.778). The STAS rates were 0.567 (95% CI, 0.478–0.652) and 0.446 (95% CI, 0.392–0.501) in the solid and papillary predominant subgroups, respectively. Additionally, the STAS rates of the lepidic, acinar, mucinous, cribriform, and colloid-predominant subgroups were 0.128 (95% CI, 0.092–0.175), 0.352 (95% CI, 0.312–0.394), 0.278 (95% CI, 0.169–0.42), 0.365 (95% CI, 0.337–0.394), and 0.167 (95% CI, 0.010–0.806), respectively.

### 3.3. Correlation between STAS and Clinicopathological Characteristics in NSCLC

Differences in clinicopathological characteristics between patients with and without STAS were investigated through a meta-analysis. NSCLCs with STAS were significantly more correlated with frequent visceral pleural, venous, and lymphatic invasions than those without STAS (Table 3). In NSCLCs with STAS, the estimated rates of visceral pleural, venous, and lymphatic invasions were 0.322 (95% CI, 0.275–0.373), 0.301 (95% CI, 0.251–0.356), and 0.391 (95% CI, 0.325–0.461), respectively. In addition, STAS is frequently observed in male patients. However, there were no significant differences in age, smoking history, tumor size, and tumor location between patients with and without STAS.

The correlations between genetic alterations and the presence of STAS were investigated in NSCLC. Patients with STAS were significantly more correlated with higher *ALK* mutations and *ROS1* rearrangement than those without STAS (Table 4). The estimated rates of *ALK* mutation and *ROS1* rearrangement in patients with STAS were 0.125 (95% CI, 0.102–0.152) and 0.040 (95% CI, 0.023–0.068), respectively. The estimated rates of *ALK* mutation and *ROS1* rearrangement in patients without STAS were 0.027 (95% CI, 0.011–0.067) and 0.009 (95% CI, 0.004–0.020), respectively. However, there were no significant differences between *EGFR* mutations and *KRAS* mutations between patients with and without STAS.

### 3.4. Prognosis of NSCLC with STAS

Patients with STAS had worse overall and recurrence-free survival (HR, 2.119; 95% CI, 1.811–2.480 and HR, 2.372; 95% CI, 2.018–2.788, respectively) (Figure 2 and Figure 3; Table 5). In the ADC subgroup, patients with STAS were significantly correlated with worse overall and recurrence-free survival (HR, 2.093; 95% CI, 1.756–2.496 and HR, 2.633; 95% CI, 2.145–3.232, respectively). In the SCC subgroup, patients with STAS had worse overall and recurrence-free survival (HR, 4.208; 95% CI, 2.190–8.083 and HR, 1.610; 95% CI, 1.066–2.431, respectively).

## 4. Discussion

Although the concept of STAS was introduced in 2015, it is not included as a diagnostic parameter in daily practice [2]. Because the presence of STAS is correlated with the prognosis and histological subtype of the patient, a detailed evaluation is needed in the pathological examination. However, despite many studies, the diagnostic criteria in daily practice are unclear. Therefore, the meta-analysis may be appropriate to help understand the clinicopathological impacts of STAS. Through this meta-analysis, we obtained the following results: (1) the estimated rate of STAS was 0.368 (95% CI, 0.336–0.401) in NSCLC; (2) STAS was frequently found in the micropapillary and solid predominant subtype; (3) STAS was significantly correlated with visceral pleural, venous, and lymphatic invasion; and (4) patients with STAS had worse overall and recurrence-free survival than those without STAS.

STAS was first defined by Kadota et al. in 2015 [2]. They reported that STAS is the identification of tumor cells that spread in the air spaces of the lung parenchyma adjacent to the edge of the tumor [2]. To evaluate the presence of STAS, the lung parenchyma adjacent to the edge of the tumor must be included in the pathological examination. The identification of STAS can be performed on the histological examination of lung cancer. In addition, the differentiation between tumor cells and other cells within the air space is not easy. Because the evaluation of STAS can be different from that of pathologists, obvious criteria are necessary for daily practice. Kadota et al. introduced the morphological patterns of tumor cells of STAS: (1) micropapillary structures; (2) solid nests of tumor islands; and (3) scattered discohesive single cells [2]. These patterns can easily differ from the lepidic growth patterns. Kadota et al. reported that the presence of STAS was correlated with lepidic, papillary, micropapillary, and solid patterns [2]. In our results, the estimated STAS rate for the lepidic subtype was the lowest among the ADC subtypes (0.128; 95% CI, 0.092–0.175). If the tumor component is a pure lepidic subtype, the actual rate of STAS can be lower than our results. The criterion for a major histopathological subtype of ADC is >5% of the overall tumor. Because the pure histological subtype of ADC is rare, differentiation between the components of STAS and tumors can be difficult. As described above, the morphological patterns of subtypes with low STAS rates, such as lepidic, acinar, and mucinous subtypes, are different from the morphological patterns of STAS. In our results, the estimated rates of STAS ranged from 12.8% to 71.9%, according to the ADC subtypes. The micropapillary subtype showed the highest STAS rate among the ADC subtypes (0.719; 95% CI, 0.652–0.778). In evaluating STAS, artificial spreading features should be distinguished from true STAS. Contamination on sectioning tissue and paraffin block is issued in pathologic examination. Especially in lung resection specimens, the possibility of the displacement of tumor cells may frequently be present along the plane of sectioning by a knife [54,55]. Lee et al. described that three tumor slices were observed under the microscope to avoid confusion with artificially detached cells [25].

STAS has been correlated with aggressive clinical features and a worse prognosis. However, due to the different diagnostic criteria and populations, conclusive information is unclear. Therefore, a meta-analysis is useful for obtaining conclusive information. In this study, STAS was significantly correlated with visceral pleural, venous, and lymphatic invasion. However, there was no significant correlation between STAS and age, smoking history, tumor size, and tumor location. Although previous studies have reported a correlation between STAS and clinicopathological characteristics, the detailed information between studies is different. Kadota et al. reported that STAS was significantly correlated with lymphovascular invasion and histological subtypes [2]. A previous study showed a correlation between STAS and the tumor site and the stage of lymph nodes [7]. In addition, they reported that STAS was significantly higher in micropapillary growth patterns than in other histological patterns [7]. However, there were no statistically significant differences between the presence of STAS and histological subtypes. STAS is more frequently found in the right lower lobe than in the left lower lobe [7]. In this study, no significant differences were observed in STAS rates between upper/middle and lower lobes (*p* = 0.078 in the meta-regression test).

Kadota et al. reported that STAS was not correlated with visceral pleural invasion [2], unlike in our results. In addition, they divided the patients into limited and lobectomy resection groups. In their study, no significant differences in the visceral pleural invasion were observed according to the presence of STAS. In addition, they suggested that STAS is a risk factor for locoregional recurrence. In patients with limited resection, the evaluation of the presence of STAS is difficult due to the insufficient inclusion of the adjacent parenchyma. In addition, the impact of the fixation method of inflation on the presence of STAS is unclear. Kadota et al. reported different prognostic impacts between the limited and lobectomy groups [2]. STAS was significantly correlated with worse recurrence-free survival in the limited resection group but not in the lobectomy resection group. However, in studies by Bains and Ren, a prognostic impact was found in both the limited and lobectomy resection groups [4,34]. We suggest that upon detecting STAS, close observation or adjuvant therapy is recommended. Cumulative studies for the necessity of further treatments will be needed in patients with STAS. Interestingly, the prognostic roles of STAS were different from those of ADC. The prognostic implications of STAS between stages I and III were not different. There was a significant difference in prognosis between patients with and without STAS in stage I but not in stage III (data not shown). Based on our results, the evaluation of STAS according to the histological subtype can be useful to predict the prognosis of the patient.

We compared the clinicopathological parameters between the STAS and non-STAS subgroups, unlike the previous meta-analysis. In addition, in this study, we showed the results using the estimated rate, but not the odds ratio between the STAS and non-STAS subgroups. Yin et al. reported a correlation between computer tomography and histological STAS in lung ADC [56]. In addition, Eguchi et al. reported the therapeutic effect of surgical treatment in T1 ADC with STAS [57]. Wang et al. demonstrated the prognostic implications of STAS in NSCLC [57]. In the meta-analysis by Liu et al., 12 eligible studies were included [58]. They studied reports from 2015 to 2018. Chen et al. studied using 14 eligible studies [59]. In Wang’s meta-analysis, the number of eligible studies included was eight [58]. Among the eight studies, the reports for ADC and SCC were six and two, respectively [58]. A total of 47 eligible articles were included. In addition, because 33 articles published after 2019 were included, the interest and importance of STAS are gradually increasing. Therefore, our results can be updated and reliable. In addition, we investigated STAS rates in various subtypes of NSCLC and compared clinicopathological characteristics between the STAS and non-STAS subgroups. However, in Wang’s report, information can be obtained based on tumor subtype, unlike our results [58]. In addition, unlike previous meta-analyses, the estimated STAS rates were investigated according to various subgroups.

Unlike previous meta-analyses, our study evaluated the differences in genetic alterations between NSCLC with and without STAS. From our results, detailed information on the clinicopathological characteristics of patients with and without STAS can be useful for the interpretation of patients with STAS. NSCLC with STAS had frequent *ALK* mutations and ROS1 rearrangement compared to NSCLC without STAS. In lung ADC, *EGFR* mutations are found more frequently in the micropapillary pattern [60,61]. However, there was no significant correlation between STAS and *EGFR* mutations or *KRAS* mutations. Understanding these genetic alterations in STAS may be important for the interpretation of molecular analyses of lung ADC.

There were some limitations to the current meta-analysis. First, the detailed criteria for STAS in NSCLC are unclear. STAS is within air spaces in the lung parenchyma beyond the edge of the main tumor, based on the definition of Kadota’s report [2]. However, the definitive distance was not defined for the edge of the tumor in most eligible studies. Han and Shiono’s reports described the distance as 0.5 mm and 0.25 mm, respectively [13,36]. However, subgroup analysis based on diagnostic criteria could not be performed due to insufficient information. Second, information based on mixed histological patterns could not be obtained from the eligible studies. Third, a detailed evaluation between STAS and distant metastasis could not be performed due to insufficient information. Fourth, a detailed investigation of the morphological patterns of STAS based on the histological subtypes of NSCLC could not be performed. Fourth, evaluating STAS grades may be needed because the extent and amount of STAS can affect a patient’s prognosis. However, it is difficult to assess due to insufficient information from eligible studies. Further evaluation for grading STAS will be needed. Fifth, another limitation concerned the lack of prospective studies for investigating STAS in the included eligible studies. Sixth, the single cell type of STAS was-not prognostic [62]. STAS is composed of three morphologic categories. However, it could not be compared the prognostic differences between morphological categories of STAS.

## 5. Conclusions

In conclusion, our results showed that STAS is frequently detected as a histological feature, as 36.8% of NSCLC cases. In addition, among adenocarcinomas, STAS is frequently found in the micropapillary and solid predominant subtypes. STAS was significantly correlated with aggressive tumor behavior and a worse prognosis. The recognition of STAS in daily practice is useful to predict the prognosis of the patient.

## Figures and Tables

**Figure 1 diagnostics-12-01112-f001:**
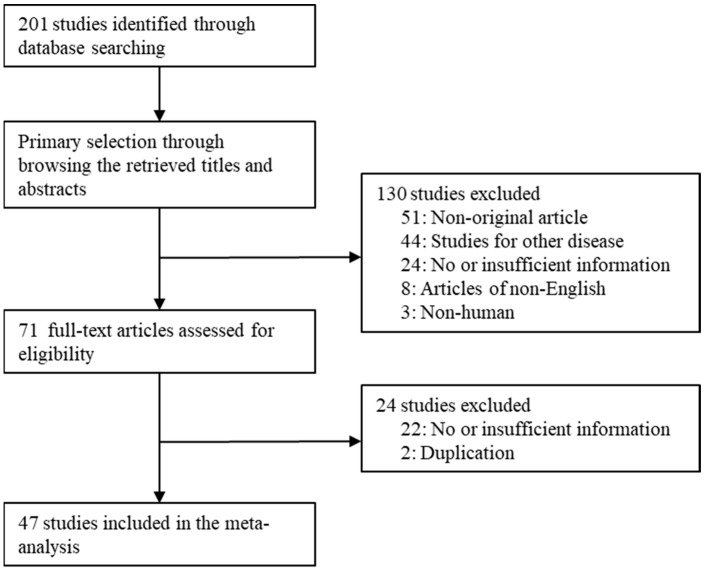
Flow chart of study search and selection methods.

**Figure 2 diagnostics-12-01112-f002:**
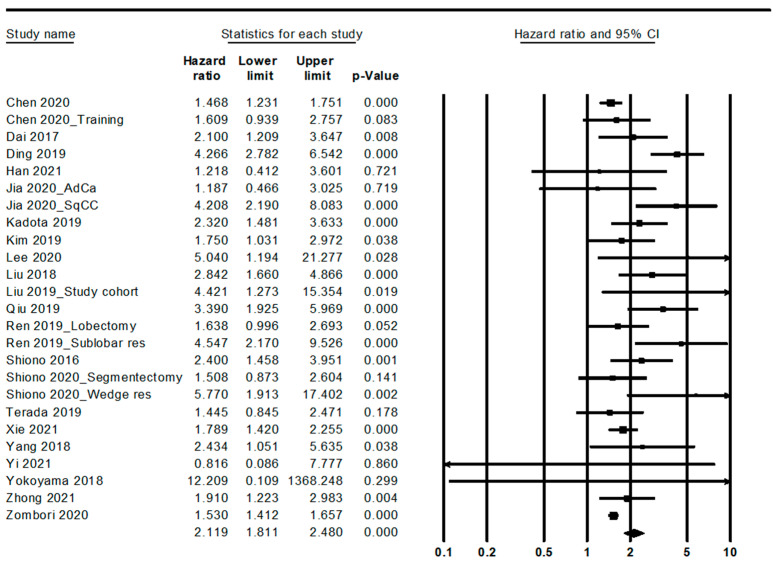
Forest plots for the overall survival.

**Figure 3 diagnostics-12-01112-f003:**
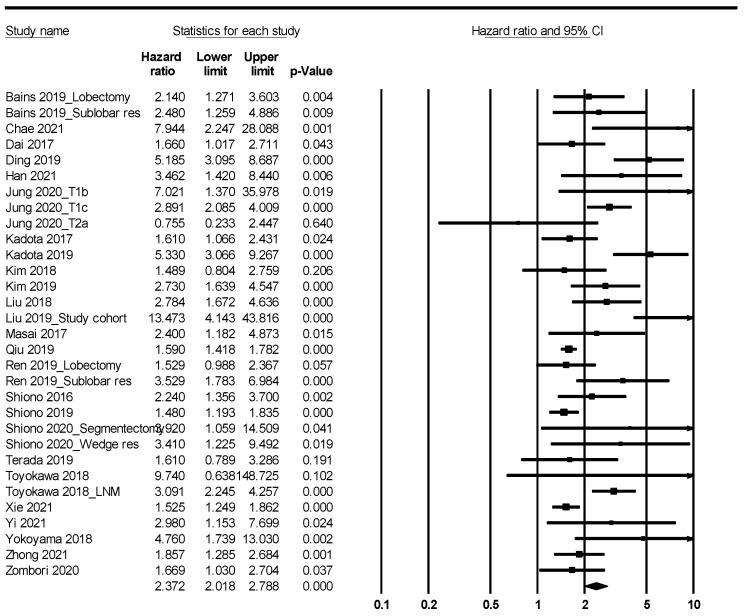
Forest plots for the recurrence-free survival.

**Table 1 diagnostics-12-01112-t001:** Main characteristics of eligible studies.

Author, Year	Location	Subtype	TNM Stage	Subgroup	No of Patients	STAS	
						Present	Absent
Alvarez Moreno 2021 [7]	USA	NSCLC	I-III		240	67	173
Bains 2019 [4]	USA	ADC	I	Lobectomy	557	191	366
		ADC	I	Sublobar resection	352	126	226
Chae 2021 [8]	Korea	ADC	I		115	20	95
Chen 2020 [9]	China	ADC	I		3346	1082	2264
Chen 2020 [10]	China	ADC	I	Training cohort	233	69	164
		ADC	I	Validation cohort	112	50	62
Dai 2017 [11]	China	NSCLC	I		383	116	267
Ding 2019 [12]	China	ADC	I-III		208	107	101
Han 2021 [13]	Korea	NSCLC	I-IV	NSCLC	1869	765	1104
		ADC	I-IV	ADC	1544	684	860
Hara 2019 [14]	Japan	ADC	I		108	32	76
Hu 2018 [15]	Taiwan	ADC	I-III		500	134	366
Ikeda 2021 [16]	Japan	NSCLC	I-III		636	282	354
Jia 2020 [17]	China	ADC	I-IV	ADC	303	183	120
		SCC	I-IV	SCC	121	39	82
Jung 2020 [18]	Korea	ADC	I		506	204	302
Kadota 2015 [2]	Japan	ADC	I		411	155	256
Kadota 2017 [19]	Japan	SCC	I-IV		216	87	129
Kadota 2019 [20]	Japan	ADC	I-IV		735	247	488
Kim 2018 [21]	Korea	ADC	I-III		276	92	184
Kim 2019 [22]	Korea	ADC	I-III		301	154	147
Kimura 2020 [23]	Japan	ADC	ND		164	29	135
Koezuka 2019 [24]	Japan	ADC	I-III		64	18	46
Lee 2018 [25]	Korea	ADC	I-III		316	160	156
Lee 2020 [26]	Korea	ADC	I-III		119	86	33
Liu 2018 [27]	China	ADC	I-III		208	107	101
Liu 2019 [28]	China	ADC	I-III	Study cohort	289	143	146
			I-III	Validation cohort	91	50	41
Lu 2017 [29]	USA	SCC	I-III		445	132	313
Masai 2017 [30]	Japan	NSCLC	ND		508	76	432
Nakajima 2021 [31]	Japan	ADC	I-III		1057	384	673
Qi 2021 [32]	China	ADC	ND		190	47	143
Qiu 2019 [33]	China	ADC	I-III		192	107	85
Ren 2019 [34]	China	ADC	I	Lobectomy	634	182	452
			I	Sublobar resection	118	43	75
Shiono 2016 [35]	Japan	ADC	I		318	47	271
Shiono 2019 [36]	Japan	NSCLC	I		848	139	709
Shiono 2020 [37]	Japan	ADC	I		217	34	183
Song 2019 [38]	China	ADC	I		277	86	191
Terada 2019 [39]	Japan	ADC	III		76	46	30
Toyokawa 2018 [40]	Japan	ADC	I		82	31	51
Toyokawa 2018 [41]	Japan	ADC	II-III	Lymph node metastasis	63	46	17
Vaghjiani 2020 [42]	USA	ADC	I-III		809	350	459
Villalba 2021 [43]	USA	ADC	I		100	43	57
Xie 2021 [44]	China	NSCLC	I-IV		803	433	370
Yang 2018 [45]	China	ADC	I		242	81	161
Yi 2021 [46]	Korea	ADC	I-II		109	41	68
Yokoyama 2018 [47]	Japan	NSCLC	I-III		35	21	14
Zhang 2020 [48]	China	ADC	I-III		762	83	679
Zhong 2021 [49]	China	ADC	I		620	167	453
Zhuo 2020 [50]	China	ADC	ND		212	107	105
Zombori 2020 [51]	Hungary	ADC	I		292	123	169

ND, no description; STAS, spread through air space; NSCLC, non-small cell lung cancer; ADC, adenocarcinoma; SCC, squamous cell carcinoma.

**Table 2 diagnostics-12-01112-t002:** Meta-analysis for the rate of spread through air space in non-small cell lung carcinoma.

	Number of Subset	Fixed Effect [95% CI]	Heterogeneity Test [*p*-Value]	Random Effect [95% CI]	Egger’s Test
Overall	53	0.367 [0.361, 0.374]	<0.001	0.368 [0.336, 0.401]	0.905
Squamous cell carcinoma	3	0.331 [0.299, 0.365]	0.025	0.338 [0.273, 0.411]	0.735
Adenocarcinoma	43	0.366 [0.358, 0.373]	<0.001	0.374 [0.340, 0.409]	0.599
Lepidic predominant	28	0.167 [0.151, 0.183]	<0.001	0.128 [0.092, 0.175]	0.126
Acinar predominant	28	0.361 [0.347, 0.374]	<0.001	0.352 [0.312, 0.394]	0.699
Papillary predominant	28	0.434 [0.414, 0.454]	<0.001	0.446 [0.392, 0.501]	0.559
Micropapillary predominant	25	0.647 [0.614, 0.679]	<0.001	0.719 [0.652, 0.778]	0.004
Solid predominant	28	0.465 [0.440, 0.491]	<0.001	0.567 [0.478, 0.652]	0.073
Mucinous predominant	7	0.282 [0.190, 0.397]	0.222	0.278 [0.169, 0.421]	0.654
Cribriform predominant	3	0.365 [0.337, 0.394]	0.605	0.365 [0.337, 0.394]	0.642
Colloid predominant	1	0.167 [0.010, 0.806]	1.000	0.167 [0.010, 0.806]	-

CI, Confidence interval.

**Table 3 diagnostics-12-01112-t003:** Comparisons of clinicopathological parameters between lung cancers with STAS and non-STAS.

	Number of Subset	Fixed Effect [95% CI]	Heterogeneity Test [*p*-Value]	Random Effect [95% CI]	Egger’s Test [*p*-Value]	Meta-Regression Test [*p*-Value]
Age (mean)						
STAS	25	66.2 [66.0, 66.4]	<0.001	63.8 [61.6, 65.9]	0.088	0.653
Non-STAS	25	68.1 [68.0, 68.2]	<0.001	63.0 [60.4, 65.4]	0.032	
Gender (Male)						
STAS	44	0.533 [0.521, 0.545]	<0.001	0.546 [0.514, 0.578]	0.298	0.008
Non-STAS	44	0.489 [0.480, 0.497]	<0.001	0.484 [0.451, 0.516]	0.748	
Current/ex-Smoking						
STAS	39	0.465 [0.452, 0.478]	<0.001	0.475 [0.418, 0.532]	0.951	0.236
Non-STAS	39	0.422 [0.412, 0.431]	<0.001	0.426 [0.369, 0.486]	0.862	
Tumor size (cm)						
STAS	20	1.91 [1.90, 1.92]	<0.001	2.45 [2.21, 2.69]	0.175	0.092
Non-STAS	20	1.65 [1.64, 1.65]	<0.001	2.99 [2.52, 3.46]	0.112	
Location (upper/middle lobe)						
STAS	11	0.646 [0.621, 0.671]	0.079	0.648 [0.612, 0.682]	0.722	0.078
Non-STAS	11	0.702 [0.684, 0.719]	0.003	0.691 [0.658, 0.721]	0.021	
Visceral pleural invasion						
STAS	30	0.355 [0.341, 0.370]	<0.001	0.322 [0.275, 0.373]	0.187	<0.001
Non-STAS	30	0.202 [0.193, 0.212]	<0.001	0.177 [0.128, 0.239]	0.478	
Venous invasion						
STAS	23	0.352 [0.335, 0.370]	<0.001	0.301 [0.251, 0.356]	0.093	<0.001
Non-STAS	23	0.151 [0.140, 0.163]	<0.001	0.120 [0.080, 0.175]	0.319	
Lymphatic invasion						
STAS	20	0.495 [0.476, 0.514]	<0.001	0.391 [0.325, 0.461]	0.005	<0.001
Non-STAS	20	0.192 [0.180, 0.205]	<0.001	0.130 [0.092, 0.181]	0.103	

CI, Confidence interval; STAS, spread through air space.

**Table 4 diagnostics-12-01112-t004:** Comparisons of genetic mutation between lung cancers with STAS and non-STAS.

	Number of Subset	Fixed Effect [95% CI]	Heterogeneity Test [*p*-Value]	Random Effect [95% CI]	Egger’s Test [*p*-Value]	Meta-Regression Test [*p*-Value]
*ALK* mutation						
STAS	7	0.125 [0.102, 0.152]	0.504	0.125 [0.102, 0.152]	0.894	<0.001
Non-STAS	7	0.042 [0.030, 0.059]	<0.001	0.027 [0.011, 0.067]	0.120	
*EGFR* mutation						
STAS	13	0.464 [0.439, 0.489]	<0.001	0.438 [0.373, 0.506]	0.421	0.058
Non-STAS	13	0.519 [0.500, 0.538]	<0.001	0.523 [0.473, 0.573]	0.864	
*ROS1* rearrangement						
STAS	3	0.040 [0.023, 0.067]	0.359	0.040 [0.023, 0.068]	0.050	0.003
Non-STAS	3	0.008 [0.004, 0.018]	0.315	0.009 [0.004, 0.020]	0.966	
*KRAS* mutation						
STAS	3	0.059 [0.039, 0.089]	0.168	0.053 [0.029, 0.096]	0.161	0.284
Non-STAS	3	0.033 [0.020, 0.053]	0.301	0.033 [0.019, 0.056]	0.375	

CI, Confidence interval; STAS, spread through air space.

**Table 5 diagnostics-12-01112-t005:** Comparisons of prognosis between lung cancers with STAS and non-STAS.

	Number of Subset	Fixed Effect [95% CI]	Heterogeneity Test [*p*-Value]	Random Effect [95% CI]	Egger’s Test [*p*-Value]
*Overall survival*	25	1.684 [1.584, 1.791]	<0.001	2.119 [1.811, 2.480]	0.001
Adenocarcinoma	21	1.656 [1.552, 1.766]	<0.001	2.093 [1.756, 2.496]	0.005
Squamous cell carcinoma	1	4.208 [2.190, 8.083]	1.000	4.208 [2.190, 8.083]	-
*Recurrence-free survival*	31	1.888 [1.763, 2.023]	<0.001	2.372 [2.018, 2.788]	<0.001
Adenocarcinoma	25	2.028 [1.869, 2.200]	<0.001	2.633 [2.145, 3.232]	<0.001
Squamous cell carcinoma	1	1.610 [1.066, 2.431]	1.000	1.610 [1.066, 2.431]	-

CI, Confidence interval; STAS, spread through air space.

## Data Availability

No new data were created or analyzed in this study. Data sharing is not applicable to this article.

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
