# Peer review of "Clinicopathological Impact of the Spread through Air Space in Non-Small Cell Lung Cancer: A Meta-Analysis"

_diagnostics, 2022, doi:10.3390/diagnostics12051112_

Round 1
Reviewer 1 Report
Thank you for the chance you gave me to read this interesting study entitled “Clinicopathological impact of the spread through air space in non-small cell lung cancer: A meta-analysis” by Pyo et al.
In this meta-analysis, the authors aimed to elucidate the clinical significance of spread through air space (STAS) in non-small cell lung cancer (NSCLC) through a meta-analysis and using 47 relevant eligible studies. Based on the published results, STAS seems to be frequently detected as a histological feature as well as to be significantly correlated with aggressive tumor behavior and a worse prognosis. This is a very interesting topic based on the significance of STAS in NSCLC and this meta-analysis provides useful conclusions. In addition, the study is well written and according to my point of view it would satisfy the appropriate criteria for publication in this journal after reducing the high similarity rate (29%) of your study based on the Turnitin.
In this meta-analysis, the authors investigated the association of spread through air spaces (STAS), which is defined as tumor cells within air spaces in the lung parenchyma beyond the edge of the main tumor, with clinicopathological characteristics of NSCLC patients as well as the clinical outcome.
There other meta-analyses with the same topic and there are some concerns regarding the novelty of the study, however, the current study, according to the authors, is based on the analysis of 47 relevant eligible studies and this is the most recent. According to my point of view, the most important issue with this study is the high similarity rate that I have mentioned in my initial comments.
In addition, the authors should also modify some terms like "clinicopathological significance" or the phrase "the presence of STAS is useful to predict the clinicopathological characteristics and prognosis of patients with NSCLC" which are misleading.
As I mentioned, there are other meta-analyses with the same topic, however, the current study, according to authors, is based on the analysis of 47 relevant eligible studies and this is the most recent.
Methodology is described well and a "flow chart" is also provided.
The authors could also provide their analysis regarding the prognostic value of the STAS taking into consideration the genetic alterations as co-factors.
The presented results are compatible with those of other studies, further confirming the negative prognostic value of STAS in NSCLC.
The authors should improve the "Introduction" section providing more data. However, the core of published studies are there.
Forest plots regarding the association of STAS with overall and recurrence-free survivals in different studies could also been provided.
In addition, a table with subgroup analyses of the associations between STAS and survivals would be very helpful (e.g.
Author Response
Thank you for the chance you gave me to read this interesting study entitled “Clinicopathological impact of the spread through air space in non-small cell lung cancer: A meta-analysis” by Pyo et al.
In this meta-analysis, the authors aimed to elucidate the clinical significance of spread through air space (STAS) in non-small cell lung cancer (NSCLC) through a meta-analysis and using 47 relevant eligible studies. Based on the published results, STAS seems to be frequently detected as a histological feature as well as to be significantly correlated with aggressive tumor behavior and a worse prognosis. This is a very interesting topic based on the significance of STAS in NSCLC and this meta-analysis provides useful conclusions. In addition, the study is well written and according to my point of view it would satisfy the appropriate criteria for publication in this journal after reducing the high similarity rate (29%) of your study based on the Turnitin.
In this meta-analysis, the authors investigated the association of spread through air spaces (STAS), which is defined as tumor cells within air spaces in the lung parenchyma beyond the edge of the main tumor, with clinicopathological characteristics of NSCLC patients as well as the clinical outcome.
There other meta-analyses with the same topic and there are some concerns regarding the novelty of the study, however, the current study, according to the authors, is based on the analysis of 47 relevant eligible studies and this is the most recent. According to my point of view, the most important issue with this study is the high similarity rate that I have mentioned in my initial comments.
Response:
As a point-out, the similarity rate based on the Turnitin could be high. The authors have previously published a large number of meta-analysis papers. Therefore, this research using meta-analysis of the same method will inevitably have some similarity.
In addition, the authors should also modify some terms like "clinicopathological significance" or the phrase "the presence of STAS is useful to predict the clinicopathological characteristics and prognosis of patients with NSCLC" which are misleading.
Response:
As a point-out, we corrected the terms in the revised manuscript.
As I mentioned, there are other meta-analyses with the same topic, however, the current study, according to authors, is based on the analysis of 47 relevant eligible studies and this is the most recent.
Response:
As a point-out, because the concept of STAS was introduced in 2015, the current study included the recent eligible studies.
Methodology is described well and a "flow chart" is also provided.
The authors could also provide their analysis regarding the prognostic value of the STAS taking into consideration the genetic alterations as co-factors.
The presented results are compatible with those of other studies, further confirming the negative prognostic value of STAS in NSCLC.
The authors should improve the "Introduction" section providing more data. However, the core of published studies are there.
Response:
As a recommendation, we added the comments using more data in the revised manuscript.
Forest plots regarding the association of STAS with overall and recurrence-free survivals in different studies could also been provided. In addition, a table with subgroup analyses of the associations between STAS and survivals would be very helpful (e.g.
Response:
As a recommendation, we added the forest plots (Fig. 2 and 3) in the revised manuscript.

Reviewer 2 Report
The authors present a meta-analysis of STAS in NSCLC with correlation to various clinicopathological factors including survival. The analysis includes a large number of studies and data on molecular alterations. The topic is relevant as STAS has gained a lot of interest but today does not alter how patients are treated.
I have some comments.
The authors conclude (line 272) that “The recognition of STAS in daily practice is useful to predict the prognosis of the patient.”
Why is this useful since the impact seem limited and it does not alter treatment?
Do the authors suggest that complementary lobectomy should be performed in cases with sublobar resection if STAS is found? (Lines 226-228)
Or is STAS a strong enough factor to suggest adjuvant therapy in small surgically resected NSCLC cases?
Also related, the authors demonstrate that STAS is linked to pleural, venous, and lymphatic invasion as well as high-grade (micropapillary and solid) growth patterns.
It would be relevant to know if STAS is to be considered an independent prognostic marker.
I think the readers would benefit from some expansion of “First, the detailed criteria for STAS in NSCLC are unclear.” (Line 261) How do criteria differ between studies?
Do the authors think STAS should be graded? E.g. PMID: 33747923 show that single cell STAS is not prognostic, suggesting that only micropapillary clusters and solid nests would count or be classified as “high-grade” STAS. There may be other suggestions such as number of clusters or how far they are from the tumor front (although I currently can not find a reference).
I don’t think an elaborate discussion on STAS as invasion pattern vs. artifact (e.g. knife-induced) is needed, but maybe the topic should be mentioned to further recognize areas of complication and provide the reader with references (e.g. well-described in PMID: 26927715 and 31943337).
(Also, manipulation of the tissue by the surgeon would be a possible cause of STAS, and I have heard surgeons confirming that significant squeezing of the tissue may occur, but I have not seen any publication mentioning this.)
Lack of prospective studies investigating STAS should be mentioned as a limitation.
Minor comments:
Most keywords are found in the title and thus do not contribute to indexing/searching.
Instead of “non-small cell lung cancer, spread through air space, and meta-analysis” maybe consider e.g. growth pattern, sublobar resection, adenocarcinoma, squamous cell carcinoma.
For the first sentence of the Introduction (line 30) the reference is not in brackets [].
On line 61 a start parenthesis ( is missing.
Author Response
The authors present a meta-analysis of STAS in NSCLC with correlation to various clinicopathological factors including survival. The analysis includes a large number of studies and data on molecular alterations. The topic is relevant as STAS has gained a lot of interest but today does not alter how patients are treated.
I have some comments.
The authors conclude (line 272) that “The recognition of STAS in daily practice is useful to predict the prognosis of the patient.”
Why is this useful since the impact seem limited and it does not alter treatment?
Do the authors suggest that complementary lobectomy should be performed in cases with sublobar resection if STAS is found? (Lines 226-228)
Or is STAS a strong enough factor to suggest adjuvant therapy in small surgically resected NSCLC cases?
Response:
Based on our results, the necessary of additional resection or adjuvant therapy can’t be evaluated. Cumulative studies for additional treatments according to detecting STAS are needed in revising guidelines. We added the comments in the revised manuscript as below:
We suggest that the detecting STAS is recommended for the close observation or additional treatment. Cumulative studies for the necessary of further treatments will be needed in patients with STAS.
Also related, the authors demonstrate that STAS is linked to pleural, venous, and lymphatic invasion as well as high-grade (micropapillary and solid) growth patterns.
It would be relevant to know if STAS is to be considered an independent prognostic marker.
Response:
Whether STAS is an independent prognostic marker could not be evaluated through a meta-analysis. The multivariate analysis will be needed to confirm this point.
I think the readers would benefit from some expansion of “First, the detailed criteria for STAS in NSCLC are unclear.” (Line 261) How do criteria differ between studies?
Response:
We added the different criteria between studies in the revised manuscript as below:
STAS was within air spaces in the lung parenchyma beyond the edge of the main tumor based on the definition of Kadota’s report [2]. However, the definitive distance was not defined for the edge of the tumor in most eligible studies. Han’s and Shiono’s reports described the distance as 0.5 mm and 0.25 mm, respectively [13,36].
Do the authors think STAS should be graded? E.g. PMID: 33747923 show that single cell STAS is not prognostic, suggesting that only micropapillary clusters and solid nests would count or be classified as “high-grade” STAS. There may be other suggestions such as number of clusters or how far they are from the tumor front (although I currently can not find a reference).
Response:
STAS is composed of three morphologic categories. The prognostic implications of STAS extent or amount or histologic type can be different. However, it could not be compared the prognostic differences between morphological categories of STAS. As a recommendation, we added the comment and references in the revised manuscript as below:
Sixth, Xie et al. reported that single cell STAS was not prognostic [44]. STAS is composed of three morphologic categories. However, it could not be compared the prognostic differences between morphological categories of STAS.
Reference
- Xie, H.; Su, H.; Zhu, E.; Gu, C.; Zhao, S.; She, Y.; Ren, Y.; Xie, D.; Zheng, H.; Wu, C.; Dai, C.; Chen, C. Morphological Subtypes of Tumor Spread Through Air Spaces in Non-Small Cell Lung Cancer: Prognostic Heterogeneity and Its Underlying Mechanism. Front. Oncol. 2021, 11, 608353.
I don’t think an elaborate discussion on STAS as invasion pattern vs. artifact (e.g. knife-induced) is needed, but maybe the topic should be mentioned to further recognize areas of complication and provide the reader with references (e.g. well-described in PMID: 26927715 and 31943337).
(Also, manipulation of the tissue by the surgeon would be a possible cause of STAS, and I have heard surgeons confirming that significant squeezing of the tissue may occur, but I have not seen any publication mentioning this.)
Response:
As a recommendation, we added the comment and references in the revised manuscript as below:
In evaluating STAS, artificial spreading features should be distinguished from the true STAS. Contamination on sectioning tissue and paraffin block is issued in pathologic examination. Especially in lung resection specimens, the possibility of displacement of tumor cells may frequently be present along the plane of sectioning by a knife [54,55]. Lee et al. described that three tumor slices were observed under the microscope to avoid confusion with artificially detached cells [25].
References
- Shih, A.R.; Mino-Kenudson, M. Updates on spread through air spaces (STAS) in lung cancer. Histopathology 2020, 77, 173-180.
- Thunnissen, E.; Blaauwgeers, H.; de Cuba, E.; Yick, C.Y.; Flieder, D.B. Ex Vivo Artifacts and Histopathologic Pitfalls in the Lung. Arch. Pathol. Lab. Med. 2016, 140, 212-220.
Lack of prospective studies investigating STAS should be mentioned as a limitation.
Response:
As a recommendation, we added the limitation in the revised manuscript as below:
Fifth, it had a limitation the lack of prospective studies for investigating STAS in the included eligible studies.
Minor comments:
Most keywords are found in the title and thus do not contribute to indexing/searching.
Instead of “non-small cell lung cancer, spread through air space, and meta-analysis” maybe consider e.g. growth pattern, sublobar resection, adenocarcinoma, squamous cell carcinoma.
Response:
As a recommendation, we corrected the keywords.
For the first sentence of the Introduction (line 30) the reference is not in brackets [].
Response:
As a recommendation, we corrected the revised version.
On line 61 a start parenthesis ( is missing.
Response:
As a recommendation, we corrected the revised version.
Round 2
Reviewer 2 Report
The authors have made proper changes and responses to suggestions. I have no further comments.